# Prognostic Markers of Ocrelizumab Effectiveness in Multiple Sclerosis: A Real World Observational Multicenter Study

**DOI:** 10.3390/jcm11082081

**Published:** 2022-04-07

**Authors:** Roberta Lanzillo, Antonio Carotenuto, Elisabetta Signoriello, Rosa Iodice, Giuseppina Miele, Alvino Bisecco, Giorgia Teresa Maniscalco, Leonardo Sinisi, Felice Romano, Maria Di Gregorio, Luigi Lavorgna, Francesca Trojsi, Marcello Moccia, Mario Fratta, Nicola Capasso, Raffaele Dubbioso, Maria Petracca, Antonio Luca Spiezia, Antonio Gallo, Martina Petruzzo, Marcello De Angelis, Simona Bonavita, Giacomo Lus, Gioacchino Tedeschi, Vincenzo Brescia Morra

**Affiliations:** 1Department of Neurosciences, Reproductive and Odontostomatological Sciences, Federico II University, 80131 Naples, Italy; robertalanzillo@libero.it (R.L.); carotenuto.antonio87@gmail.com (A.C.); rosa.iodice@unina.it (R.I.); moccia.marcello@gmail.com (M.M.); nicolacapasso91@gmail.com (N.C.); rafdubbioso@gmail.com (R.D.); maria@petraccas.it (M.P.); antonio.luca.spiezia@hotmail.com (A.L.S.); martinapetruzzo@gmail.com (M.P.); marcello.deangelis91@gmail.com (M.D.A.); vincenzo.bresciamorra2@unina.it (V.B.M.); 2Department of Advanced Medical and Surgical Sciences, II Clinic of Neurology, University of Campania Luigi Vanvitelli, 80131 Naples, Italy; elisabetta.signoriello@gmail.com (E.S.); giuseppinamiele20@gmail.com (G.M.); mario.fratta@unicampania.it (M.F.); simona.bonavita@unicampania.it (S.B.); giacomo.lus@unicampania.it (G.L.); 3Department of Advanced Medical and Surgical Sciences, University of Campania Luigi Vanvitelli, 80138 Naples, Italy; alvino.bisecco@unicampania.it (A.B.); francesca.trojsi@unicampania.it (F.T.); antonio.gallo@unicampania.it (A.G.); gioacchino.tedeschi@unicampania.it (G.T.); 4Multiple Sclerosis Center, Antonio Cardarelli Hospital, 80131 Naples, Italy; gtmaniscalco@libero.it; 5Neurological Unit, San Paolo Hospital, ASL Napoli 1 Centro, 80125 Naples, Italy; leosinisi@libero.it; 6Neurological and Stroke Unit, CTO Hospital, AORN Ospedali dei Colli, 80131 Naples, Italy; felrom@alice.it; 7Medical Sciences Department, Azienda Ospedaliero-Universitaria San Giovanni di Dio e Ruggi d’Aragona, 84125 Salerno, Italy; mariadigregorio82@gmail.com; 8Department of Human Neurosciences, Sapienza University, 00185 Rome, Italy

**Keywords:** progression, multiple sclerosis, ocrelizumab, disease-modifying treatment, real-world

## Abstract

Pivotal trials showed the effectiveness of the monoclonal antibody ocrelizumab in relapsing and progressive multiple sclerosis (MS). However, data on everyday practice in MS patients and markers of treatment effectiveness are scarce. We aimed to collect real-world data from ocrelizumab-treated MS patients, relapsing-remitting (RR) and progressive MS patients (PMS), including active secondary progressive MS (aSPMS) and primary progressive MS (PPMS) patients, and to explore potential prognostic factors of clinical outcome. Patients were enrolled at MS centres in the Campania region, Italy. We collected clinic-demographic features retrospectively one year before ocrelizumab start (T_−1_), at ocrelizumab start (T_0_), and after one year from ocrelizumab start (T_1_). We explored possible clinical markers of treatment effectiveness in those patients receiving ocrelizumab treatment for at least one year using multilevel-mixed models. We included a total of 383 MS patients (89 RRMS and 294 PMS; 205 females, mean age: 45.8 ± 11.2, disease duration: 12.7 ± 11.6 years). Patients had a mean follow-up of 12.4 ± 8.2 months, and 217 patients completed one-year ocrelizumab treatment. Overall, EDSS increased from T_−1_ to T_0_ (coeff. = 0.30, 95% coefficient interval [CI] = 0.19–0.41, *p* < 0.001) without a further change between T_0_ and T_1_ (*p* = 0.61). RRMS patients did not show an EDSS change between T_−1_ and T_0_ nor between T_0_ and T_1_. Conversely, PMS patients showed EDSS increase from T_−1_ to T_0_ (coeff. = 0.34, 95% CI = 0.22–0.45, *p* < 0.001) without a further change between T_0_ and T_1_ (*p* = 0.21). PMS patients with a time from conversion shorter than 2 years showed increased EDSS from T_−1_ to T_0_ (coeff. = 0.63, 95% CI = 0.18–1.08, *p* = 0.006) without a further change between T_0_ and T_1_ (*p* = 0.94), whereas PMS patients with a time from conversion longer than 2 years showed increased EDSS from T_0_ to T_1_ (coeff. = 0.30, 95% CI = 0.11–0.49, *p* = 0.002). Naïve patients showed an EDSS decrease between T_0_ and T_1_ (coeff. = −0.30, 95% CI = −0.50–−0.09, *p* = 0.004). In conclusion, our study highlighted that early ocrelizumab treatment is effective in modifying the disability accrual in MS patients.

## 1. Introduction

Ocrelizumab (OCR) is a humanized monoclonal antibody that acts by depleting CD20+ B cells while preserving innate immunity. OCR was reported to be effective on disease inflammatory activity and progression for both relapsing-remitting (RR), active progressive and primary progressive multiple sclerosis (MS) patients in the OPERA I/II [1,2] and ORATORIO [3] randomised clinical trials (RCTs) without major safety concerns. Similarly, OCR was reported to prevent disability accrual in both RRMS and primary progressive MS patients and hence, it may also be useful in those patients with progressive disease course with or without relapses following the clear-cut RR disease stage (i.e., active secondary progressive MS) [4,5,6]. 

While RCTs are extremely useful to assess treatment safety and efficacy, the population included in these studies do not necessarily reflect the heterogeneity of the population in clinical settings. Hence, real-world data are always warranted to confirm treatment efficacy and safety in clinical settings [7]. Since its approval, OCR has been widely used in clinical practice, and few real-world studies have confirmed its safety and efficacy profile in either monocentric [4,8,9,10,11] or multicentric [5,12,13] settings. The aforementioned studies and previous post-hoc analyses from RCTs also sought to investigate possible predictors of OCR efficacy with conflicting results. Specifically, post-hoc analyses from RCTs showed that OCR was equally effective in all patients independently of their age, sex, previous disease-modifying treatment and baseline EDSS [6,14].

In contrast, in a monocentric real-world setting, naïve patients and patients with a lower EDSS benefited the most from the OCR effect in terms of disability accrual [4]. Studies evaluating the predictors of OCR effectiveness in a real-world, multicentric framework are still lacking. In addition, neither the post-hoc analysis from RCTs nor the data from real-world studies explored the association between the OCR treatment start in relation to time from confirmed progressive clinical phenotype diagnosis to OCR effectiveness. In line with previously published data [15,16], we hypothesized that OCR would show an effectiveness profile overlapping with data from RCTs and that OCR may be more effective in preventing disability accrual when introduced upon the appearance of the course of the progressive disease.

## 2. Methods

### 2.1. Procedures and Participants

This retrospective study included MS patients starting treatment with OCR according to clinical practice between January 2018 and December 2019 at nine MS centres in the Campania region of Italy. We performed a descriptive analysis for the whole study sample and an analysis of effectiveness, including those patients with at least 1 year of follow-up. For each patient, we collected the following information: demographic data (i.e., gender and age), history of MS (i.e., MS onset date, MS course (RRMS or active secondary progressive and primary progressive following Lublin’s criteria [17]), annual relapse rate (ARR) in the 2 years before OCR start, time from conversion to progressive MS and EDSS 1 year before OCR start when available), date of OCR start, previous disease-modifying treatment (DMTs (DMTs were classified as a first-line treatment for interferon, glatiramer acetate, teriflunomide, dimethyl fumarate or as a second-line treatment for natalizumab, rituximab, siponimod, fingolimod and alemtuzumab)), EDSS at OCR start, and clinical outcomes (i.e., EDSS 1 year after OCR start and number of MS relapses). The time from conversion to secondary progressive MS was retrospectively collected. Clinicians made the SPMS diagnosis at each participating site following the Lublin criteria [17] based on a ‘steadily increasing, objectively documented neurological dysfunction/disability without recovery’. Patients were classified as naïve if they did not receive any DMTs before OCR started. 

The present study was conducted in accordance with specific national laws and the ethical standards of the 1964 Declaration of Helsinki and its later amendments. Given its retrospective design, this study did not interfere with the care received by patients. In addition, specific ethical approval was not required due to the retrospective design and since all clinical assessments were part of the clinical practice in a university- or hospital-based specialized centre setting. However, as per Italian regulations [18], the principal investigator site notified the local ethics committee ‘Carlo Romano’ about this retrospective study. Patients provided their informed consent to collect data for clinical purposes.

### 2.2. Statistical Analyses

Demographic and clinical features for the study population are presented using mean standard deviation (SD), median and range as appropriate. Statistical analysis was performed by AC, who was blind to patients’ identities and did not contribute to data collection. The effectiveness of OCR was evaluated by analysing (i) occurrence of relapse, (ii) EDSS at 1 year after OCR start. To properly evaluate the effect of OCR in modifying the trajectory of disability accrual, we evaluated changes in EDSS 1 year before OCR start (T_−1_), EDSS at OCR start (T_0_), and EDSS 1 year after OCR start (T_1_) using multivariable mixed models including time points as factor of interest, EDSS as the dependent variable, age, gender and centre as covariates, and subject ID as a random factor. To further explore possible demographic and clinical features affecting the effectiveness of OCR in modifying disability accrual trajectories, following the previous post-hoc analysis from RCTs [6], we performed the same analysis by dividing patients according to age (less or more than 40 years), gender, clinical disease course (relapsing vs. progressive MS), progressive MS course (active secondary progressive MS vs. primary progressive), DMTs status (naïve vs. non-naïve) and EDSS (<4 vs. ≥4). 

Furthermore, we also performed the same analysis by dividing patients according to disease duration (≤10 years vs. >10 years) and time from secondary progressive MS diagnosis following Lublin’s phenotypic classification [17] (≤2 years vs. >2 years). Cut-offs were selected considering that 10 years is the time from relapsing-remitting to progressive disease course in terms of natural MS history [19,20]. In comparison, 2 years is the transitioning period where uncertainty exists for clinicians in defining the transition from relapsing to progressive disease course [21].

## 3. Results

### 3.1. Clinico-Demographic Features

We included 383 OCR-treated patients (205 females, mean age 45.8 ± 11.2 years). Eighty-nine patients (23%) were RRMS, whereas 294 patients (77%) were progressive MS (PMS) patients. Specifically, 165 patients (43.1%) were active secondary progressive patients and 129 (33.7%) were primary progressive MS patients. Median EDSS was 5.5 (1–8.5) with a median disease duration of 11 years (0–41). Seventy-four patients (19.3%) were naïve to any DMTs, 154 patients (40.2%) switched to OCR from first-line DMTs, and 155 patients (40.5%) switched to OCR from second-line DMTs. Overall, patients were followed up for 12.4 ± 8.2 months, with 217 (57%) patients receiving follow-up for more than 12 months. Demographic and clinical features for the total sample and for the sample included in the effectiveness analysis are reported in Table 1.

### 3.2. Effectiveness Analysis

Patients included in the effectiveness analysis did not differ in terms age, sex, ARR, disease duration, EDSS at T_−1_ and T_0_ and treatment status compared with the whole sample. Conversely, in the effectiveness analysis, there was a higher prevalence of both active secondary progressive MS patients (43% vs. 46%, *p* = 0.04) and primary progressive MS patients compared with the whole sample (34% vs. 39%, *p* = 0.04) (Table 1). For those patients who received follow-up for at least 1 year, two out of 185 (1%) patients experienced a relapse over the follow-up time period with a mean time from T_0_ of 5 months.

Overall, EDSS increased from T_−1_ to T_0_ (coeff. = 0.30, 95% coefficient interval [CI] = 0.19–0.41, *p* < 0.001) without a further change between T_0_ and T_1_ (*p* = 0.61) (Figure 1).

### 3.3. Post-Hoc Effectiveness Analysis

The results of the post-hoc analysis are summarised in Table 2 and Figure 2. Age and gender did not affect OCR effectiveness.

Patients with a disease duration shorter than 10 years showed increased EDSS from T_−1_ to T_0_ (coeff. = 0.43, 95% CI = 0.24–0.63, *p* < 0.001) without a further change between T_0_ and T_1_ (*p* = 0.23). Conversely, patients with disease duration longer than 10 years showed increased EDSS from T_−1_ to T_0_ (coeff. = 0.21, 95% CI = 0.08–0.34, *p* = 0.002) with a trend toward a further EDSS increase from T_0_ and T_1_ (coeff. = 0.13, 95% CI = 0.00–0.27, *p* = 0.05).

Patients with EDSS at T_0_ lower than 4 showed no EDSS change between T_−1_, T_0_ and T_1,_ while patients with EDSS at T_0_ higher than 4 showed increased EDSS from T_−1_ to T_0_ (coeff. = 0.36, 95% CI = 0.25–0.47, *p* < 0.001) without a further change between T_0_ and T_1_ (*p* = 0.65).

Non-naïve patients showed increased EDSS from T_−1_ to T_0_ (coeff. = 0.28, 95%CI = 0.17–0.40, *p* < 0.001) without a further change between T_0_ and T_1_ (*p* = 0.10). Naïve patients showed EDSS decrease between T_0_ and T_1_ (coeff. = −0.30, 95%CI = −0.50–−0.09, *p* = 0.004).

RRMS patients did not show EDSS change between T_−1_ and T_0_ nor between T_0_ and T_1_. Conversely, PMS patients showed EDSS increase from T_−1_ to T_0_ (coeff. = 0.34, 95%CI = 0.22–0.45, *p* < 0.001) without a further change between T_0_ and T_1_ (*p* = 0.21). In detail, active secondary progressive MS patients showed EDSS increase from T_−1_ to T_0_ (coeff. = 0.23, 95%CI = 0.12–0.35, *p* < 0.001) and between T_0_ and T_1_ (coeff. = 0.19, 95%CI = 0.05–0.33, *p* = 0.009) while primary progressive MS patients showed EDSS increase from T_−1_ to T_0_ (coeff. = 0.34, 95%CI = 0.21–0.46, *p* < 0.001) without a further change between T_0_ and T_1_ (*p* = 0.87). When grouping secondary progressive MS patients according to time from conversion (shorter or longer than 2 years), only patients with a shorter time from conversion showed stable EDSS during therapy (shorter than 2 years: coeff. = 0.02, 95%CI = −0.42–0.45, *p* = 0.94; longer than 2 years: coeff. = 0.30, 95%CI = 0.11–0.49, *p* = 0.002)

## 4. Discussion

Previous studies have shown that DMTs are generally more effective in RRMS if used earlier in the disease course than later [16]. OCR was also reported to be more effective in preventing the reaching of the milestone EDSS 6.5 when started early over the disease course in RRMS [15]. Similarly, it may be hypothesized that also in progressive clinical phenotypes, early OCR treatment start might show a higher efficacy in mitigating disease activity and disability accrual. In light of this background, in this study, we aimed to investigate the efficacy of OCR treatment in both RRMS and PMS patients in a real-world, multicentric Italian setting and to explore the potential impact of early OCR introduction as a prognostic factor for treatment success in reducing disability accrual.

The clinical effectiveness data in our cohort are generally in line with those reported in the pivotal phase 3 clinical trials and in the few real-world studies conducted so far [2,3,8,9,10,11,12].

With patients who received follow up for at least 1 year, OCR showed a strong effect on relapse occurrence, with only two patients experiencing relapses with a mean time from T_0_ of 5 months. Compared with the OCR phase 3 trials OPERA I and II, our RRMS cohort at the time of OCR initiation was older (37 vs. 46 years), had a longer disease duration (6.7 vs. 11 years) and patients were less frequently treatment-naïve (74% vs. 19%) [2]. The mean number of relapses in the year prior to OCR initiation was remarkably lower than in the OPERA trials (mean ± SD, 0.28 ± 0.33 vs. 1.3 ± 0.6) [2]. This may be explained by the older age, longer disease duration, lower proportion of treatment-naïve patients in our cohort and by the requirement of relapse activity prior to screening as a key eligibility criterion in the OPERA trials [2]. Despite these differences, OCR efficacy on clinical disease activity was remarkable.

Overall, in our study, EDSS did not change with OCR treatment despite a previously increasing trend from T_−1_ to T_0_, in line with OCR efficacy on EDSS progression shown both in RR and primary progressive MS patients in OPERA and ORATORIO studies [2,3]. According to the OPERA and ORATORIO results, we also reported OCR efficacy in slowing disability accrual in the RRMS and primary progressive MS patients at the subgroup analysis in our real-world setting. This would suggest that OCR could not exert any effect on disability accrual on secondary progressive MS. However, taking a closer look at our post-hoc effectiveness analysis revealed that patients with time from conversion to secondary progressive MS ≤ 2 years benefited the most from OCR treatment in terms of disability accrual after 1 year of follow-up. In addition, DMT-naïve patients showed an EDSS decrease between T_0_ and T_1_, thus suggesting that, independently of disability level and disease duration, OCR is more able to exert an effect on an immune system, which has not been previously targeted with other medications [22]. Therefore, despite the short follow-up period, our results yield evidence of a precocious positive action of OCR on disability accrual trajectory also in secondary progressive MS patients, since the trend was of EDSS increase over the previous year.

These are relevant data helping MS experts in unravelling therapeutic decisions since we now have evidence of outcome prognostic factors.

The Limitations of our study include the retrospective design, which does not consider all possible confounding factors. Also, our population is from a well-defined geographical region, and its generalizability cannot be fully addressed. In addition, the follow-up is limited to 12.4 ± 8.2 months. This framework may hamper the proper evaluation of OCR efficacy for slowing down EDSS progression, and hence, real-world studies with periods of longer follow-up should be performed to confirm our results. In addition, the high variability in the sample clinical features (i.e., large disease duration and EDSS span) may also impact the significance of our results. Unfortunately, real-world studies are burdened with a large variability of demographic and clinical features for subjects involved in the study, especially in a multicentric framework. While this could be regarded as a limitation, such variability reflects the clinical settings and hence, represents the way clinicians may choose among different treatments to efficaciously treat their patients. Finally, we based our study on clinical outcomes. MRI outcomes are of utmost importance in multiple sclerosis. However, these data could not have been retrieved in a standardised manner in our multicentric study; hence, we could not draw any conclusion about OCR efficacy in terms of MRI activity in MS.

## 5. Conclusions

Our study confirmed in a real-world setting that OCR is an effective treatment for both RRMS and PMS patients. Furthermore, OCR is more effective in patients at the early stages during the disease course (naïve patients and patients recently converting to progressive phenotypes). Our study highlighted that early OCR treatment is effective in modifying the disability accrual in MS patients and, hence, OCR should be administered timely to catch the therapeutic window.

## Figures and Tables

**Figure 1 jcm-11-02081-f001:**
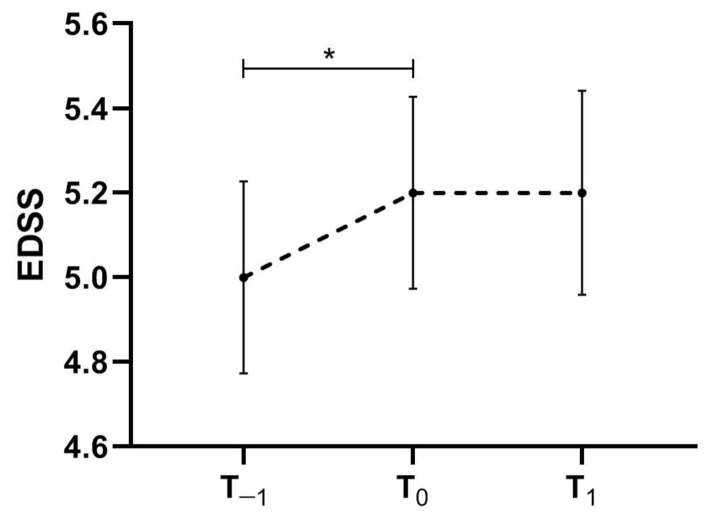
EDSS trajectory between T_−1_, T_0_ and T_1_ for the whole multiple sclerosis sample included in the effectiveness analysis. * *p* < 0.05 at multivariable mixed models including time-points as factor of interest, EDSS as the dependent variable, age, gender and centre as covariates and subject ID as a random factor. Dash line represents EDSS trajectory.

**Figure 2 jcm-11-02081-f002:**
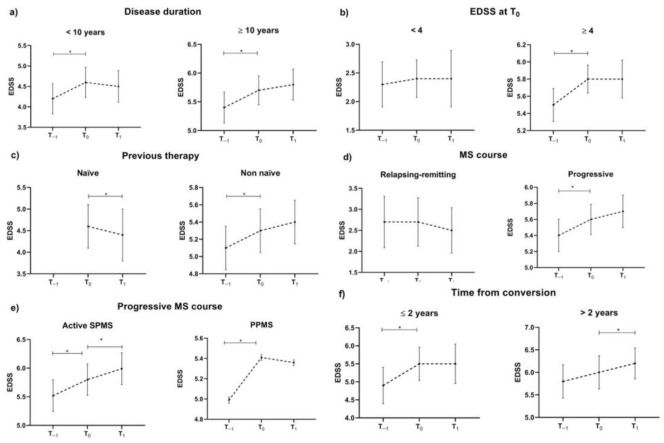
Post-hoc effectiveness analysis. EDSS trajectory between T_−1_, T_0_ and T_1_ according to disease duration (**a**), EDSS at T_0_ (**b**), treatment status (**c**), MS disease course (**d**,**e**) and time from conversion (**f**). * *p* < 0.05 at multivariable mixed models including time-points as a factor of interest, EDSS as the dependent variable, age, gender and centre as covariates and subject ID as a random factor. Dash lines represent EDSS trajectory.

**Table 1 jcm-11-02081-t001:** Demographic and clinical features for the whole sample and for patients included in the effectiveness analysis.

	Total Population	Patients with at Least 1 Year Follow-Up	*p*-Value
Number of subjects	383	217	
Sex			
Male, N (%)	178 (46.5)	115 (53)	0.13
Female, N (%)	205 (53.5)	102 (47)	
Age, mean (SD) (years)	45.8 (11.2)	46.6 (10.6)	0.37
ARR pre-OCR start, mean (SD)	0.29 (0.3)	0.28 (0.33)	0.72
EDSS 1 year pre-OCR start, median (Range)	5 (0–8)	5.25 (1–8)	0.39
EDSS at OCR start, median (Range)	5.5 (1–8.5)	5.5 (1–8.5)	0.27
Disease duration, median (Range) (years)	11 (0–41)	11 (1–41)	0.76
MS course			
Relapsing-remitting, N (%)	89 (23)	32 (15)	0.04 *
Active secondary progressive, N (%)	165 (43)	100 (46)	
Primary progressive, N (%)	129 (34)	85 (39)	
Previous Therapy			
First-line, N (%)	154 (40.2)	96 (44)	0.78
Second-line, N (%)	155 (40.5)	80 (37)	
Naïve, N (%)	74 (19.3)	41 (19)	
Ocrelizumab courses, median (Range)	2 (1–5)	4 (2–5)	<0.001 *

Abbreviations: N = number; SD = standard deviation; MS = multiple sclerosis; ARR = annualised relapse rate; OCR = ocrelizumab, EDSS = expanded disability status scale. * whole sample vs. sample for the effectiveness analysis, *p* < 0.05.

**Table 2 jcm-11-02081-t002:** Post-hoc effectiveness analysis.

Features	Number	EDSS (Mean [SD])				
		T_−1_	T_0_	T_1_	*p*-Value (T_0_ vs. T_−1_)	*p*-Value (T_1_ vs. T_0_)
Sex						
Male	115	4.9 (1.8)	5.1 (1.7)	5.2 (1.8)	<0.001 *	0.08
Female	102	5.1 (1.7)	5.3 (1.6)	5.2 (1.8)	<0.001 *	0.26
Age						
<40 years	63	4.0 (2.0)	4.3 (2.0)	4.5 (2.2)	<0.001 *	0.10
≥40 years	154	5.3 (1.4)	5.5 (1.4)	5.5 (1.5)	<0.001 *	0.78
Disease duration						
<10 years	94	4.2 (1.8)	4.6 (1.8)	4.5 (1.9)	<0.001 *	0.23
≥10 years	123	5.4 (1.5)	5.7 (1.4)	5.8 (1.5)	0.002 *	0.05
EDSS at T_0_						
<4	38	2.3 (1.2)	2.4 (1.0)	2.4 (1.5)	0.96	0.88
≥4	179	5.5 (1.3)	5.8 (1.1)	5.8 (1.5)	<0.001 *	0.65
Previous Therapy						
Naïve	41	-	4.6 (1.6)	4.4 (1.9)	-	0.004 *
Non-naïve	176	5.1 (1.7)	5.3 (1.7)	5.4 (1.7)	<0.001 *	0.09
MS course						
Relapsing-remitting	32	2.7 (1.7)	2.7 (1.6)	2.5 (1.5)	0.60	0.08
Progressive	185	5.4 (1.4)	5.6 (1.3)	5.7 (1.4)	<0.001 *	0.21
Progressive MS course						
Active secondary progressive MS	100	5.5 (1.4)	5.8 (1.4)	6.0 (1.4)	<0.001*	0.009*
Primary progressive MS	85	5.0 (0.1)	5.4 (0.1)	5.4 (0.1)	<0.001*	0.87
Time from conversion						
≤2 years	24	4.9 (1.2)	5.5 (1.1)	5.5 (1.3)	0.006 *	0.94
>2 years	66	5.8 (1.5)	6.0 (1.5)	6.2 (1.4)	0.10	0.002 *

Abbreviations: MS = multiple sclerosis; EDSS = expanded disability status scale. * multivariable mixed models including time-points as a factor of interest, EDSS as the dependent variable, age, gender and centre as covariates and Subject ID as a random factor.

## Data Availability

Data will be made available upon reasonable request to the corresponding author.

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
