# Peer review of "Prognostic Markers of Ocrelizumab Effectiveness in Multiple Sclerosis: A Real World Observational Multicenter Study"

_jcm, 2022, doi:10.3390/jcm11082081_

Round 1
Reviewer 1 Report
RWD is always important and welcome, but in this manuscript there are some major weaknesses. Firstly, the study method is unclear for the reader. What is a retrospective analysis of prospectively collected data? It seems that no approval was obtained from the participants to take part in the study, eventhough it was prosspectively collected, but only for the data collection from the patient files. Are the treating neurologists the same as those who collected and analyzed the data? When evaluating only efficacy one would need to have MRI results. The study design as a whole is unclear and raises ethical and reliability concearns. Also the study population is confusing. In the abstract only RMS and PMS (relapsing forms of MS) were mentioned, whereas in the results a great deal of subjects were primary progressive (PPMS) patients. In the dicussion a term "progressive forms of MS" is used. Does this include both PPMS and PMS? One of the results is the time to conversion to SPMS. No criteria for SPMS is given and this result is unreliable taken in to account a very short follow up. The discussion is weak. It merely contains repetition of the results, no actual discussion or references.
Author Response
Reviewer 1
Review RWD is always important and welcome, but in this manuscript there are some major weaknesses. Firstly, the study method is unclear for the reader. What is a retrospective analysis of prospectively collected data? It seems that no approval was obtained from the participants to take part in the study, even though it was prospectively collected, but only for the data collection from the patient files. When evaluating only efficacy one would need to have MRI results. The study design as a whole is unclear and raises ethical and reliability concearns.
We have now clarified in the method section that is is a retrospective study. We have also included the following statement in the ethic section “The present study was conducted in accordance with specifc national laws and the ethical standards laid down in the 1964 Declaration of Helsinki and its later amendments. Given its retrospective design, in no way this study did interfere with the care received by patients. Due to the retrospective design and since all clinical assessments were part of clinical practice in a University or Hospital specialized Centre setting, specific ethical approval was not required.”
Are the treating neurologists the same as those who collected and analyzed the data?
Treating neurologists collected data whilst data analysis was performed from AC, who did not performed data collection. AC was blind to patients’ identity. We have now specified this in the methods section and amended in the data contribution statement
Also the study population is confusing. In the abstract only RMS and PMS (relapsing forms of MS) were mentioned, whereas in the results a great deal of subjects were primary progressive (PPMS) patients. In the dicussion a term "progressive forms of MS" is used. Does this include both PPMS and PMS?
We thank the reviewer for this comment. We have now clarified in the abstract, method and result section that we included relapsing-remitting MS and progressive MS patients. This latter label included both active secondary progressive MS and primary progressive MS.
One of the results is the time to conversion to SPMS. No criteria for SPMS is given and this result is unreliable taken in to account a very short follow up.
We thank the reviewer for this remark. Actually, SPMS diagnosis was drawn from treating neurologist according to Lublin criteria for clinical course of MS (Neurology. 2014 Jul 15; 83(3): 278–286.). We would like to highlight that we used time from SPMS diagnosis to analyse whether patients with a time from conversion shorter than 2 years show a better effectiveness from ocrelizumab treatment. Hence, this was a data collected retrospectively, We did not seek to analyse time to conversion during OCR treatment. We have now better detailed this in the method section.
The discussion is weak. It merely contains repetition of the results, no actual discussion or references.
We thank the reviewer for letting us further discussing our data. We have now discussed our data both in terms of ARR reduction and EDSS progression reduction by comparing our data with previously available findings from RCT and RWD.

Reviewer 2 Report
The authors presented a very interesting study regarding the effectiveness of ocrelizumab in a real-world setting. However, I have several issues that I would like to address.
1. short follow-up, mean of 12.4+/-8 months is not so sufficient time to evaluate effectiveness in real-world data. The authors did not include this in the limitations of the study.
2. it is not listed have all patients included in the effectiveness analysis received the ocrelizumab treatment during follow-up? What are the number and the median of ocrelizumab cycles?
3. effectiveness analysis group is driven mostly by the progressive patients, how do authors explain this ratio with a significant prevalence of progressive patients?
4. authors mentioned conversion to progressive MS? How is this conversion defined?
5. a high number of patients of the analyzed cohort are relapsing-progressive (43.1%). According to Lublin classification, these could be SPMS and PPMS patients - just as a suggestion I think it would be valuable to define groups better (RRMS, PPMS and SPMS) in order to evaluate effectiveness in different MS phenotypes. EDSS span is from 1.0-8.5 in the study cohort with a median of 5.5 indicating severely disabled patients.
Author Response
Please find attached our reply to reviewer report.

Reviewer 3 Report
Please find comments attached

Author Response
PLease find attached the reply to reviewer

Round 2
Reviewer 1 Report
Anyhow, about the article. It has major ethical problems, which cannot be solved. There is no ethical evaluation, no consent from the patients, no approval from the hospitals. The authors merely state that declaration of Helsinki was followed and that is not enough.
Methodology is also inadequate. the authors state that they collected all relapsing and active progressive patients. When they actually collected also primary progressive patients as well. The follow up is far too short and real efficacy outcomes, mainly MRI is missing.
Author Response
Please find attached our reply o review report

Reviewer 2 Report
The authors acknowledged all comments, provided explanations to all questions raised and performed analysis as suggested.
Author Response
We thank the reviewer for her/his valuable suggestions